# Transoral Robotic Approach for Pharyngeal Schwannoma

**DOI:** 10.3390/diagnostics15040484

**Published:** 2025-02-17

**Authors:** Riccardo Nocini, Valerio Arietti, Sokol Sina, Luca Sacchetto

**Affiliations:** 1Unit of Otolaryngology, Head and Neck Department, University of Verona, P.le L.A. Scuro 10, 37134 Verona, Italy; riccardo.nocini@aovr.veneto.it (R.N.); luca.sacchetto@univr.it (L.S.); 2Department of Diagnostics and Public Health, Section of Pathology, Azienda Ospedaliera Universitaria Integrata di Verona, Piazzale Aristide Stefani 1, 37126 Verona, Italy; sokol.sina@aovr.veneto.it

**Keywords:** robotic surgery, oropharynx, dysphagia, schwannomna, head and neck

## Abstract

**Introduction**: Schwannomas are a common condition encountered in clinical ENT practice, but they are rarely found in the pharyngeal or laryngeal regions. **Materials and Methods**: In this report, we share our experience using the transoral robotic approach to treat a schwannoma located on the lateral pharyngeal wall, with the surgery being performed exclusively using transoral robotic surgery (TORS). It is important to be aware of any pseudocystic lesions in this area, as they can lead to unpredictable anatomopathological outcomes. **Results and Conclusions**: Our technique allows us to achieve complete resection of the tumor, facilitating rapid recovery for the patient without the need for additional treatment.

**Figure 1 diagnostics-15-00484-f001:**
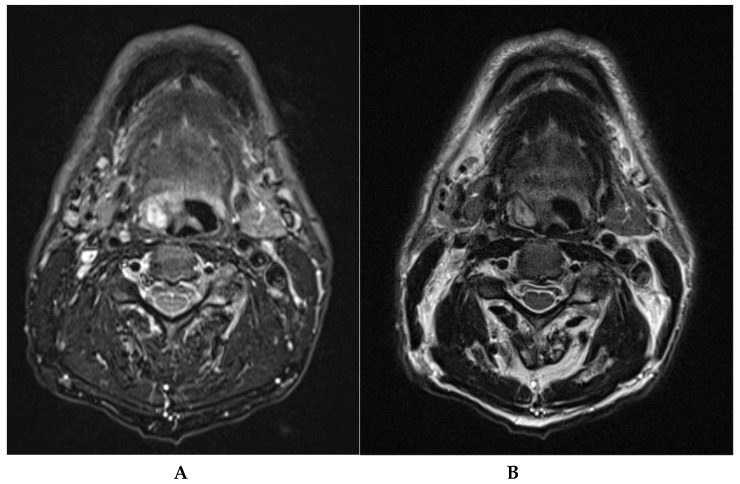
(**A**,**B**) A 75-year-old patient has been complaining for several months of slowly progressive dysphagia and a foreign body sensation in the throat when swallowing. The patient has no other relevant medical comorbidities except mild hyperparathyroidism. He was a non-smoker. Fiberoptic clinical examination revealed a 1.5 cm nodule on the right glosso-epiglottic vallecula, which appeared to be pedunculated at the base of the tongue and the lateral pharyngeal wall. The cervical lymph nodes were not palpable, and the rest of the exam was normal. The patient’s routine hematologic examination and urinalysis were normal. An MRI scan revealed a 15 × 8 mm oval nodule at the right tongue base/vallecula. It was well-defined, isointense to muscle on T1WI without contrast enhancement, and hyperintense on T2WI (Figure 1A,B). The MRI did not provide a definitive diagnosis but suggested that the lesion exhibited benign characteristics. The mass was excised under general anesthesia with the TORS technique using the Xi Da Vinci System robot (Intuitive**^®^**, Sunnyvale, CA, USA).A schwannoma is a benign, encapsulated, and slow-growing neoplasm that arises from the Schwann cells of the peripheral nerve sheath [1,2,3]. About one third of all schwannomas are found in the head and neck region, with cranial nerve VIII (vestibulocochlear nerve) being the most frequently affected nerve in this area. Intraoral occurrences are relatively rare, comprising only 1–12%, with the tongue being the most frequently affected site [4]. There are two types of nerve sheath tumors of the peripheral nerves: neurofibromas and schwannomas. Neurofibromas are composed of neurites, Schwann cells, and fibroblasts embedded within a collagenous or myxoid matrix. Schwannomas, on the other hand, derive from Schwann cells and are typically encapsulated [5,6]. There is no sex predilection and the neoplasm can occur alone without a known etiology [7,8]. They may also be associated with genetically inherited diseases, such as Neurofibromatosis type 1 (NF1) and type 2 (NF2), and schwannomatosis [9]. These tumors can develop from any nerve that is surrounded by a Schwann cell sheath. This includes the cranial nerves (except for the optic and olfactory nerves), the spinal nerves, and the autonomic nervous system [4]. Identifying the nerve of origin (hypoglossal, glossopharyngeal, and lingual) in the tongue can be challenging due to their proximity. Dreher et al. reported that in over 50% of intraoral tumors, distinguishing between tumors of these nerves is difficult [10,11]. Schwannomas of the tongue most commonly manifest between the second and fourth decades of life. These tumors are not gender-specific and typically present as a painless mass. Schwannomas can lead to significant symptoms if they are located in the posterior region of the tongue or if they reach a substantial size [4]. Symptoms are usually mild and often occur in conjunction with other conditions, such as foreign body sensation in the mouth, difficulty swallowing, and, rarely, shortness of breath or other life-threatening symptoms. The intensity of the symptoms is determined by the location and size of the tumor [12]. If the tumor is asymptomatic, it is often misdiagnosed, but if it is symptomatic, transoral resection is considered the method of choice for the treatment of the vast majority of these tumors. With the advent of transoral robotic surgery (TORS), resection has become even less invasive and more precise, as the region is often not as easily accessible (usually the base of the tongue or vallecula) and the lesion is not prone to infiltration. Therefore, the recurrence rate after excision is very low and malignant transformation is very rare [4,12,13,14]. Given the rarity of this pathology, there are few records in the literature, so the number of cases is insufficient to provide real guidelines.

**Figure 2 diagnostics-15-00484-f002:**
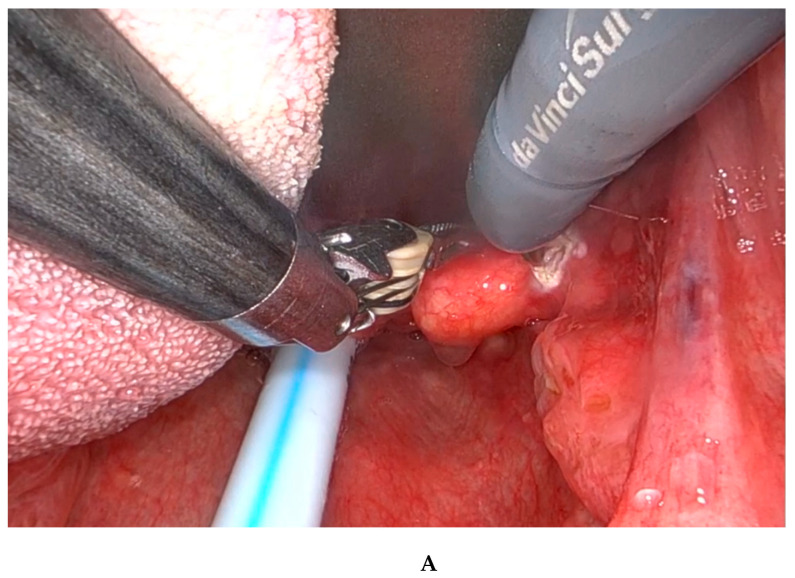
(**A**,**B**) The procedure was conducted under general anesthesia. Nasotracheal intubation was preferred, although orotracheal intubation was also an option. The patient was placed in the supine position and a Boyle–Davis oral gag was used to expose the oropharynx. In our institution, the Da Vinci robotic system (Intuitive^®^, Sunnyvale, CA, USA) was used with a rigid 0-degree endoscope. The first surgeon sat at the command station while the second surgeon sat at the patient’s head and assisted in cleaning the oropharynx and suctioning. Grasping forceps were used to gently pull the lesion and bipolar forceps (Maryland dissector) were used to cauterize and excise from the base of the tongue and the lateral pharyngeal wall, taking care to remove the lesion completely. The surgery lasted less than 30 min and there were no complications or bleeding. The patient stayed in the hospital for one night and was discharged the next day. He was instructed to eat soft and fresh food and avoid physical exertion for 15 days. Antibiotic therapy was not required. No postoperative complications were reported. The patient had no recurrence of the lesion, and 1 month after excision the dysphagia and foreign body sensation had disappeared.

**Figure 3 diagnostics-15-00484-f003:**
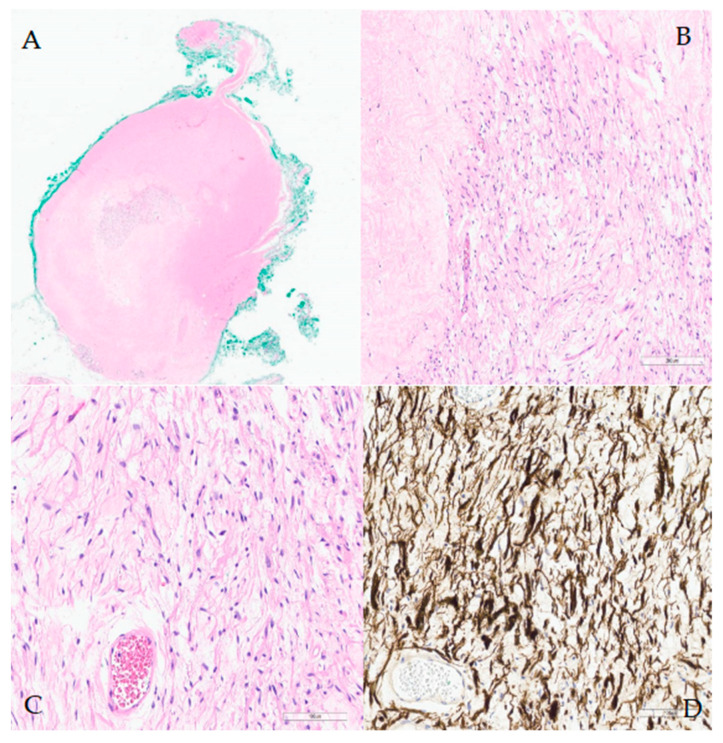
The gross examination revealed an encapsulated nodule measuring 1 cm, characterized by a moderately firm, white-pink cut surface, with no signs of ulceration. Histopathologic and immunohistochemical analyses confirmed the schwannoma diagnosis. Immunohistochemistry was positive for S-100 protein, supporting this diagnosis (Figure 3). (**A**) Low-power view of the fragment submitted, the hypocellularity is clearly visible; hematoxylin–eosin stain, 0.8×. (**B**) Low-power view of spindle cell with adjacent stromal sclerosis; hematoxylin–eosin stain, 10×. (**C**) High-power view of spindle cells without atypia; hematoxylin–eosin stain, 20×. (**D**) Positive immunohistochemistry for S100 confirmed the diagnosis of schwannoma, 20×. In the evaluation of a patient presenting with a tongue tumor, schwannoma should be included in the differential diagnosis alongside lipoma, neurofibroma, hemangioma, lymphangioma, lingual thyroid, leiomyoma, and benign salivary gland tumors [14]. Intraoral occurrences of schwannomas in the head and neck region are relatively rare, representing approximately 1–12% of cases [4]. Schwannoma is a non-cancerous, encapsulated, slow-growing, and typically solitary tumor that originates from Schwann cells of the nerve sheath. In the oral cavity, it most commonly affects the tongue, followed by the palate, floor of the mouth, buccal mucosa, gingiva, lips, and vestibular mucosa. Schwannomas affecting the tongue are typically localized in the anterior third of the tongue (about two-thirds versus one-third in the posterior two-thirds). Symptoms depend on the tumor’s location and size. Patients with small tumors, measuring less than 2 cm, located in the anterior third of the tongue, generally do not exhibit symptoms. Intraoral schwannomas commonly cause symptoms including pharyngeal discomfort, dysphagia, and changes in voice. Furthermore, when these tumors are located on the tongue, they may result in snoring or ulceration [13]. Patients with larger tumors or tumors in the two posterior thirds of the tongue are more likely to develop symptoms, as the posterior lesions can cause snoring and dysphagia [7]. Schwannomas have several defining features histologically. They typically have a capsule. The tumor alternates between hypercellular spindle cell areas (Antoni A) and hypocellular round cell areas (Antoni B). Schwannomas show strong and diffuse immunoreactivity for the S-100 protein, which aids in diagnosis [15]. The preferred imaging technique for diagnosing lingual schwannomas is Magnetic Resonance Imaging (MRI). MRI offers superior tissue contrast compared to computed tomography (CT), enabling more accurate localization and enhanced visualization of the tumor’s relationship with adjacent structures. Additionally, MRI facilitates more precise measurement of the tumor size. On MRI scans, schwannomas typically present as well-defined masses that do not infiltrate surrounding tissues [4,7]. On T1-weighted imaging, the lesions appear isointense in relation to the muscle, whereas on T2-weighted imaging they appear as hyperintense lesions [4,13]. Due to their benign nature and minimal symptomatology, intraoral schwannomas are frequently subject to misdiagnosis. Schwannomas located in the oral cavity are more readily diagnosed as patients typically notice them due to their manifestation as a mass. Conversely, lesions situated at the posterior third of the tongue, part of the oropharynx, are often misdiagnosed because of their asymptomatic nature when small and their cyst-like appearance. Consequently, the incidence of pathology might be higher. Lesions in the oral cavity are generally accessible through the classical transoral approach under direct or magnified vision and can be excised with relative ease. However, schwannomas located in the oropharynx, particularly at the base of the tongue, present a challenge due to the difficult access. In such cases, transoral robotic surgery (TORS) is increasingly becoming the preferred method for treatment when symptomatic. TORS significantly enhances surgical procedures through improved visualization and instrumentation. It offers high-resolution, three-dimensional (3D) views, allowing surgeons to navigate the patient’s mouth more effectively. With flexible telescopes and innovative wrist instruments that provide seven degrees of movement, TORS enables precise bimanual tissue manipulation, reaching areas previously difficult to access [16]. Several cost–benefit analyses need to be conducted in the future. It is well known that this procedure can be costly, and not all healthcare systems can afford it. As a result, the significant potential of this procedure remains unknown, particularly in cases of benign lesions that can typically be removed using traditional transoral methods. Additionally, the setup time, while limited in an experienced center, can still be considered a drawback. To establish sufficient statistical credibility, prospective, randomized studies with large sample sizes are necessary. This procedure is commonly performed without complications. Post-complete excision, the recurrence rate is low, and patients are typically discharged within 1 to 2 days. A cold, soft diet is recommended for the first 10–14 days, and physical activity should be avoided for two weeks to mitigate the risk of bleeding.

## Data Availability

All data used are available within this article.

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
