# Peer review of "Transoral Robotic Approach for Pharyngeal Schwannoma"

_diagnostics, 2025, doi:10.3390/diagnostics15040484_

Round 1
Reviewer 1 Report
Comments and Suggestions for Authors The case study reviews a case of a well-described tumor in an unusual location. Already established in the literature: extracranial Schwannomas are unusual but most commonly encountered in the head & neck. Standard of care treatment is complete excision. This case study makes the following contributions: Schwannomas can present in the tongue base and may be accessed via transoral robotic surgery for standard-of-care complete excision. The approach is not novel for tongue base tumors, as TORS is already used extensively for SCCa. TORS is also well-described for tongue base operations for obstructive sleep apnea and benign tongue base tumors. This case is a natural extension of those applications, and its novelty is driven by the unusual pathology. The review of Schwannoma pathology and pathophysiology is appropriate but also not novel.Which davinci system was used? Appears to be the Xi but that detail would help.
I would accept the manuscript on the grounds that it is interesting, after revisions/clarifications in my other review.
Author Response
Dear Reviewer,
We are very pleased to hear that you enjoyed our article. We appreciate your feedback and have made the requested modifications. While we acknowledge that this work is based on a single case and therefore has its limitations, we believe it contributes to the existing literature on lesions treated with TORS, which is currently limited in scope.
Thank you for your review.
Best regards,
The Editors.
Reviewer 2 Report
Comments and Suggestions for Authors
this case report concerns the excision of a small schwannoma of right glosso-epiglottic valecula with DaVinci.
Although this is manuscript offers a brief and relatively interesting review on oral schwannomas, there are many reports in the literature on the excision of various lesions of the base of the tongue and the hypopharynx with TORS. Thus I feel that this report is not of a general interest for the general or head and neck otolaryngologist.
Author Response
Dear Reviewer,
Thank you for your feedback. We acknowledge that this work is based on a single case, which has its limitations. However, we believe it contributes to the existing literature on lesions treated with Transoral Robotic Surgery (TORS), an area that is currently limited in scope. This may encourage the use of robotic surgery for benign pathologies of the head and neck, and we hope it will lead to further studies with a greater number of cases to establish more significant findings. At this time, we do not have a clear understanding of the cost-effectiveness of this procedure.
Thank you for your review.
Best regards,
The Editors.
Reviewer 3 Report
Comments and Suggestions for Authors
The authors report a case of pharyngeal schwannoma which has been removed using trans oral robotic approach. It was undoubtedly an advance way of removing the tumour. However, in order to show the superiority of the technique, the authors should also discuss on the usual approaches to excise the lesion as well as their challenges and how the robotic technique may overcome those challenges.
Author Response
Dear Reviewer,
Thank you for your feedback. As you suggested, we have added a paragraph at the end that highlights the advantages TORS offers to surgeons. However, we currently do not have a cost-effectiveness ratio, mainly due to the absence of studies demonstrating this in benign diseases. We hope to contribute to this area in the future.
Thank you for your review.
Best regards,
The Editors.
Round 2
Reviewer 1 Report
Comments and Suggestions for Authors
I thank the authors for their corrections.
Author Response
Dear reviewer,
we would like to thank you once again for your revisions.
Best regards, the authors.
Reviewer 2 Report
Comments and Suggestions for Authors
although I have initially suggested the rejection of the manuscript, the reasons that MDPI provides for rejection of a paper ((article has serious flaws, additional experiments needed, research not conducted correctly) are not applicable in this case report. Thus, I modified my initial decision, although I still believe that is not really useful for the readers, and a benign lesion in this area can be easily removed the majority of the times with the classical microlaryngoscopy. Nevertheless, I believe that at least in the discussion, the authors should provide their explanations that they gave to me (TORS may be useful for benign lesions for reasons of exposure and safety) stating at the same time the disadvantages of TORS (cost, time for the set up)
Author Response
Dear Reviewer,
Thank you once again for your valuable feedback. As you suggested, we have added a paragraph discussing the drawbacks of the procedure. The aim of this article is to demonstrate the feasibility of the procedure and to stimulate further research on the topic. While we currently do not know the utility of the procedure for benign lesions, we are actively investigating it.
Thank you again for your input.
Sincerely,
The Authors
Reviewer 3 Report
Comments and Suggestions for Authors
The manuscript has been sufficiently improved to warrant publication.
Author Response
Dear Reviewer,
we would like to thank you again for your suggestions and opinions.
Best regards,
The authors
Round 3
Reviewer 2 Report
Comments and Suggestions for Authors
dear authors
thank you for your time and your consideration